# Feedback control of Wnt signaling based on ultrastable histidine cluster co-aggregation between Naked/NKD and Axin

Melissa V Gammons[†], Miha Renko[†], Joshua E Flack[†‡], Juliusz Mieszczanek[†], Mariann Bienz*

MRC Laboratory of Molecular Biology, Cambridge Biomedical Campus, Cambridge, United Kingdom

**Abstract** Feedback control is a universal feature of cell signaling pathways. Naked/NKD is a widely conserved feedback regulator of Wnt signaling which controls animal development and tissue homeostasis. Naked/NKD destabilizes Dishevelled, which assembles Wnt signalosomes to inhibit the β-catenin destruction complex via recruitment of Axin. Here, we discover that the molecular mechanism underlying Naked/NKD function relies on its assembly into ultra-stable decameric core aggregates via its conserved C-terminal histidine cluster (HisC). HisC aggregation is facilitated by Dishevelled and depends on accumulation of Naked/NKD during prolonged Wnt stimulation. Naked/NKD HisC cores co-aggregate with a conserved histidine cluster within Axin, to destabilize it along with Dishevelled, possibly via the autophagy receptor p62, which binds to HisC aggregates. Consistent with this, attenuated Wnt responses are observed in CRISPR-engineered flies and human epithelial cells whose Naked/NKD HisC has been deleted. Thus, HisC aggregation by Naked/NKD provides context-dependent feedback control of prolonged Wnt responses.

*For correspondence:
mb2@mrc-lmb.cam.ac.uk

[†]These authors contributed
equally to this work

Present address: [‡]Mosa Meat,
Maastricht, The Netherlands

Competing interests: The
authors declare that no
competing interests exist.

Reviewing editor: Melanie
Königshoff, University of
Colorado, United States

## Introduction

Cells communicate via a handful of highly conserved signaling pathways to coordinate growth and differentiation. Signal responses are often mediated by signalosomes, dynamic protein condensates that are assembled transiently at the plasma membrane by hub proteins, to transduce incoming signals to the cytoplasm and nucleus of cells (*Bienz, 2014*; *Case et al., 2019*; *Schaefer and Peifer, 2019*). A well-studied hub protein is Dishevelled, a pivotal component within the ancient Wnt signaling pathway, which controls cell specification during development and adult tissue homeostasis throughout the animal kingdom (*Cadigan and Nusse, 1997*; *Clevers et al., 2014*). Dishevelled transduces Wnt signals through canonical or non-canonical effector branches to elicit distinct cellular readouts, whereby the best-studied one is the β-catenin-dependent canonical branch which requires interaction of Dishevelled with Axin (*Angers and Moon, 2009*; *Gammons and Bienz, 2018*).

In order to transduce Wnt signals to β-catenin, Dishevelled assembles signalosomes by dynamic head-to-tail polymerization of its DIX domain (*Schwarz-Romond et al., 2007*). This enables it to bind to the DIX domain of Axin, which leads to inhibition of the associated kinases in the β-catenin destruction complex *Stamos et al., 2014* whose assembly, in the absence of signaling, depends on polymerization by the Axin DIX domain (*Fiedler et al., 2011*). In other words, the flow through the canonical Wnt signaling pathway is determined by the opposing activities of polymerizing Axin and Dishevelled and their mutual interaction via their DIX domains (*Bienz, 2014*): the polymerization of Axin promotes destabilization of β-catenin and therefore ensures quiescence of the pathway, while the polymerization of Dishevelled allows signalosome assembly and co-polymerization with Axin,

which allows β-catenin to accumulate in the nucleus and engage in transcriptional activation of Wnt target genes (*Gammons and Bienz, 2018*). Recent high-resolution imaging studies have visualized endogenous Wnt signalosomes in mammalian cells, which result from limited polymerization of Dishevelled (*Kan et al., 2020*; *Ma et al., 2020*).

Polymerization of Dishevelled and Axin is concentration-dependent (*Schwarz-Romond et al., 2007*), and occurs spontaneously if the cellular concentration of either of these proteins rises above a critical threshold. This undermines the physiological control of the signaling flux by Wnt proteins, with detrimental consequences for cells including lethality (e.g. *Fiedler et al., 2011*; *Penton et al., 2002*). It is therefore imperative that the levels of Dishevelled and Axin are tightly controlled in receiving cells. Indeed, Dishevelled and Axin are each recognized by specific ubiquitin E3 ligases that allow their individual targeting for proteasomal degradation (*Angers et al., 2006*; *de Groot et al., 2014*; *Ji et al., 2017*; *Mund et al., 2015*; *Wei et al., 2012*; *Zhang et al., 2014*; *Zhang et al., 2011a*). In addition, Dishevelled is also targeted for degradation by autophagy via association with the autophagy receptor p62 and the ubiquitin-like proteins LC3 or GABARAPL (*Gao et al., 2010*; *Ma et al., 2015*; *Zhang et al., 2011b*). Note that any mechanism earmarking whole signalosomes for degradation could result in either termination or maintenance of Wnt responses, depending on whether the Wnt signaling flux is limited in a given receiving cell by Wnt agonists such as Dishevelled, or by Wnt antagonists such as Axin.

One of the least-understood factors promoting the degradation of Dishevelled is the feedback regulator Naked Cuticle (Naked) and its vertebrate orthologs NKD1 and NKD2 (NKD). Naked was discovered in *Drosophila* as a Wnt-inducible antagonist of the Wnt pathway (*Zeng et al., 2000*). NKD orthologs also accumulate upon Wnt stimulation of vertebrate cells, and Dishevelled was identified as a key target for downregulation by NKD (*Rousset et al., 2001*; *Van Raay et al., 2007*; *Wharton et al., 2001*; *Yan et al., 2001*). Indeed, Naked/NKD is the only known intracellular feedback regulator of Wnt signaling that is conserved throughout the animal kingdom (*Figure 1—figure supplement 1*), which argues strongly for its ubiquitous function and physiological relevance. However, Naked/NKD function is not always essential (e.g. mice can be born without Nkd proteins, albeit at submendelian ratios and with cranial bone abnormalities; *Zhang et al., 2007*), as is typical for modulatory feedback regulators, which are widespread and serve to canalize signal responses during development, and render them robust (*Freeman, 2000*).

Molecularly, Naked/NKD contains a single EF-hand, which binds to the PDZ domain of Dishevelled (*Rousset et al., 2001*; *Rousset et al., 2002*; *Wharton et al., 2001*; *Yan et al., 2001*). This results in destabilization of Dishevelled, apparently via the ubiquitin/proteasome system (*Guo et al., 2009*; *Hu et al., 2010*; *Schneider et al., 2010*). Cumulative evidence indicates that Naked/NKD affects both β-catenin-dependent and non-canonical Wnt signaling responses (*Angonin and Van Raay, 2013*; *Creyghton et al., 2005*; *Hu et al., 2010*; *Marsden et al., 2018*; *Rousset et al., 2001*; *Schneider et al., 2010*; *Van Raay et al., 2007*; *Van Raay et al., 2007*; *Wharton et al., 2001*; *Yan et al., 2001*). Given that all known Wnt responses depend on Dishevelled, this is consistent with Dishevelled being a physiological target of Naked/NKD.

Here, we examine the molecular mechanism by which Nkd1 controls Wnt responses in human epithelial cells. This critically depends on a highly conserved histidine cluster (HisC) in its C-terminus which forms ultra-stable decameric core aggregates in vitro. Nkd1 also forms HisC aggregates upon accumulation in Wnt-stimulated cells, promoted by its interaction with Dishevelled. Notably, these HisC aggregates bind selectively to Axin in vitro and in vivo, thereby promoting its destabilization in cells. We used CRISPR engineering to generate human epithelial cell lines bearing specific HisC deletions of both NKD paralogs, which compromises their ability to sustain Wnt signal transduction to β-catenin and to destabilize Axin during prolonged Wnt stimulation. Similarly, CRISPR-engineered HisC deletion of *Drosophila nkd* results in embryonic defects reflecting reduced Wingless responses. These results indicate the physiological relevance of Naked/NKD HisC in flies and human cells, and reveal cellular contexts in which Naked/NKD acts as an agonist of Wnt signaling by promoting the destabilization of Axin via HisC aggregation. We also discover the autophagy p62 receptor as a HisC-dependent binding partner of Nkd1, implicating autophagy as the underlying mechanism.

## Results

### The Nkd1 HisC is crucial for ternary complex formation with Axin and DVL2

We used co-overexpression assays in HEK293T cells, monitoring Wnt signaling with a co-transfected β-catenin-dependent transcriptional reporter (SuperTOP) (*Veeman et al., 2003*), to confirm that murine HA-Nkd1 reduces the signaling activity of co-overexpressed DVL2-GFP (a human Dishevelled paralog), but not of co-overexpressed β-catenin (*Figure 1—figure supplement 2*). Previous work established that this downregulation depends on the EF-hand of Nkd1, the Dishevelled-binding domain (*Rousset et al., 2001*; *Rousset et al., 2002*; *Wharton et al., 2001*; *Yan et al., 2001*). In addition, HA-Nkd1 is also less active in downregulating β-catenin signaling if its C-terminal HisC is deleted (ΔHisC), despite the substantially higher expression levels of this deletion mutant (*Figure 1A*). The latter is consistently observed, suggesting that HisC functions to destabilize Nkd1.

Using co-immunoprecipitation (coIP) to monitor the association between co-expressed HA-Nkd1 and DVL2-GFP, we found that HA-Nkd1 also coIPs with Flag-Axin1 upon co-expression (*Figure 1B*), as previously observed (*Miller et al., 2009*). This indicates that the three proteins can form a ternary complex. While conducting these coIP assays, we discovered that overexpressed Nkd1 forms high-molecular weight (HMW) aggregates that are resistant to boiling in sodium dodecyl sulphate (SDS), and whose formation critically depends on HisC (*Figure 1C*; we were unable to detect endogenous NKD HMW aggregates, owing to the lack of suitable antibodies). Notably, Axin1-GFP specifically co-aggregates with HMW but not LMW (low-molecular weight) Flag-Nkd1 via its own internal HisC (*Figure 1D*). Thus, our data are consistent with the notion that the simultaneous binding of Naked/NKD to Dishevelled and its co-aggregation with Axin may allow it to target Axin-containing Wnt signalosomes.

### NKD1 HisC specifically co-aggregates with the internal HisC of Axin

Next, we tested whether recombinant HisC from NKD1 (identical sequence in mouse and human) would self-aggregate in vitro, by purifying lipoyl domain (Lip)-tagged protein (Lip-NKD-HisC) following expression in bacteria. Indeed, Lip-NKD-HisC forms stable aggregates that elute as a broad peak in the 250–450 kDa range (corresponding to 15–25-mers), as monitored by size exclusion chromatography (SEC) (*Figure 2A*). If the pH is lowered below ~ 6.5, these aggregates begin to dissociate, and Lip-NKD-HisC is completely monomeric at pH 5 (*Figure 2B*). This sharp pH-dependence reflects the pKa value (~6.0) of the imidazole side-chain of His: in acidic conditions (pH < 6), this side-chain is protonated and positively charged, and aggregation is thus blocked by electrostatic repulsion, whereas at higher pH values (pH > 6.2), the majority of His side-chains are uncharged, which is permissive for NKD-HisC core formation. Most probably, the majority of the His residues in Lip-NKD-HisC have to be deprotonated for aggregation (i.e. become uncharged, achieved at pH > pKa), while the protonation of a few critical His residues (pH < pKa) triggers disassembly of NKD-HisC aggregates owing to electrostatic repulsion between positively charged histidines.

If assayed by polyacryl-amide gel electrophoresis (PAGE) under denaturing and reducing conditions (i.e. after boiling in 1% w/v SDS and 5% v/v mercapto-ethanol), an ultra-stable NKD-HisC core of ~ 150 kDa is observed. This core was sized more accurately by mixing monomeric Lip-NKD-HisC with monomeric NKD-HisC bearing a Maltose-Binding Protein tag (MBP-NKD-HisC) at different ratios at acidic pH, and by increasing the pH subsequently, to allow aggregation. Because of the different molecular masses of these tags, we observe a discrete laddering on PAGE, revealing that the stable NKD-HisC core consists of a 10-mer (*Figure 2C*). Notably, the formation of NKD-HisC aggregates not only depends on the most conserved histidines, but also on two aromatic residues within HisC that are highly conserved amongst Naked/NKD orthologs (*Figure 1—figure supplement 1*): double- or quintuple point-substitutions of the most conserved histidine or aromatic residues (m1-m3) strongly reduce aggregation of recombinant Lip-NKD-HisC (*Figure 2D*). Our results indicate that NKD-HisC aggregates into a decameric ultra-stable core (called HisC core below) that can be decorated by additional NKD-HisC monomers that are less firmly associated (*Figure 2E*).

We also tested full-length HA-Nkd1 bearing these point-substitutions in transfected HEK293T cells, for their ability to form HisC aggregates and to block β-catenin signaling by co-expressed Flag-DVL2. Indeed, the two double-point mutations (m1, m2) attenuate aggregation of HA-Nkd1 in

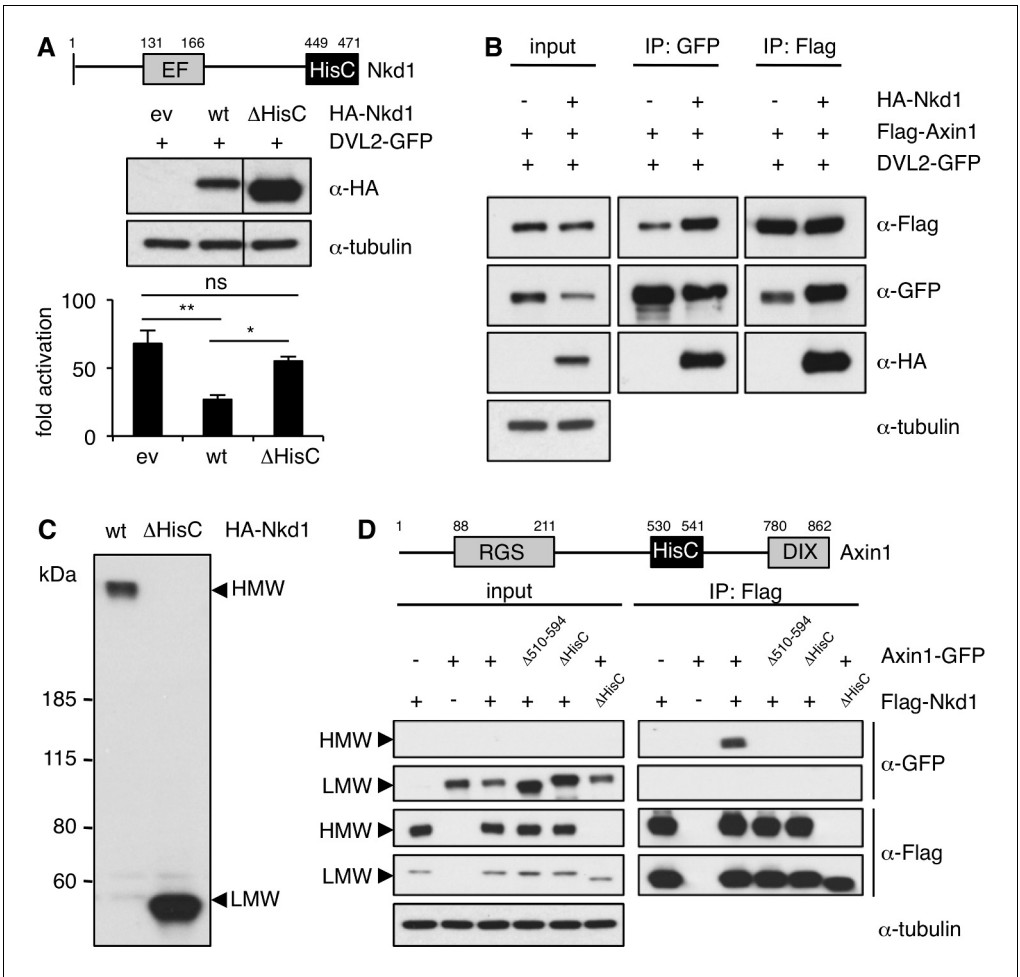

**Figure 1.** Nkd1 co-aggregates with Axin1 via highly conserved HisCs. (**A**) SuperTOP assays, monitoring the signaling activity of overexpressed DVL2-GFP upon co-expression with wt or ΔHisC HA-Nkd1 in transfected HEK293T cells (used in this and all subsequent figures unless otherwise stated); expression levels were monitored by Western blot (*above*); error bars, SEM of > 3 independent experiments; one-way ANOVA with multiple comparisons; *$p < 0.05$, **$p < 0.01$. (**B**) CoIP assays between co-expressed proteins, as indicated above panels. (**C**) Western blot of wt and ΔHisC HA-Nkd1, revealing HisC-dependent HMW aggregates; *left*, positions of molecular weight markers (LMW, corresponding to monomeric Nkd1); in subsequent figures, only HMW and LMW regions are shown (see also supplementary information, for full-length gels). (**D**) CoIP assays between co-expressed proteins, as indicated above panels, monitoring ΔHisC-dependent co-aggregation of Flag-Nkd1 and Axin1-GFP (note that the amounts of transfected DNA has been adjusted in this experiment, to obtain comparable expression levels of different constructs).

The online version of this article includes the following figure supplement(s) for figure 1:

**Figure supplement 1.** Sequence conservation of Naked/NKD orthologs.

**Figure supplement 2.** Effects of overexpressed NKD1 on β-catenin signaling.

cells, particularly noticeable when co-expressed with Flag-DVL2, while the quintuple mutation (m3) is almost as strong as ΔHisC in blocking aggregation (*Figure 2—figure supplement 1*). Notably, each of the three mutations also stabilizes HA-Nkd1, with m3 being almost as potent as ΔHisC. However, despite their significantly increased levels, none of the three mutants is as active as wt HA-Nkd1 in antagonizing Flag-DVL2 signaling: m1 reduces signaling only partially, while m2 and m3 abolish the activity of HA-Nkd1 in antagonizing Flag-DVL2 signaling (*Figure 2—figure supplement 1*). Therefore, these substitutions of highly conserved residues within HisC behave as expected from our in vitro results with recombinant HisC protein. They provide strong evidence that the HisC-dependent

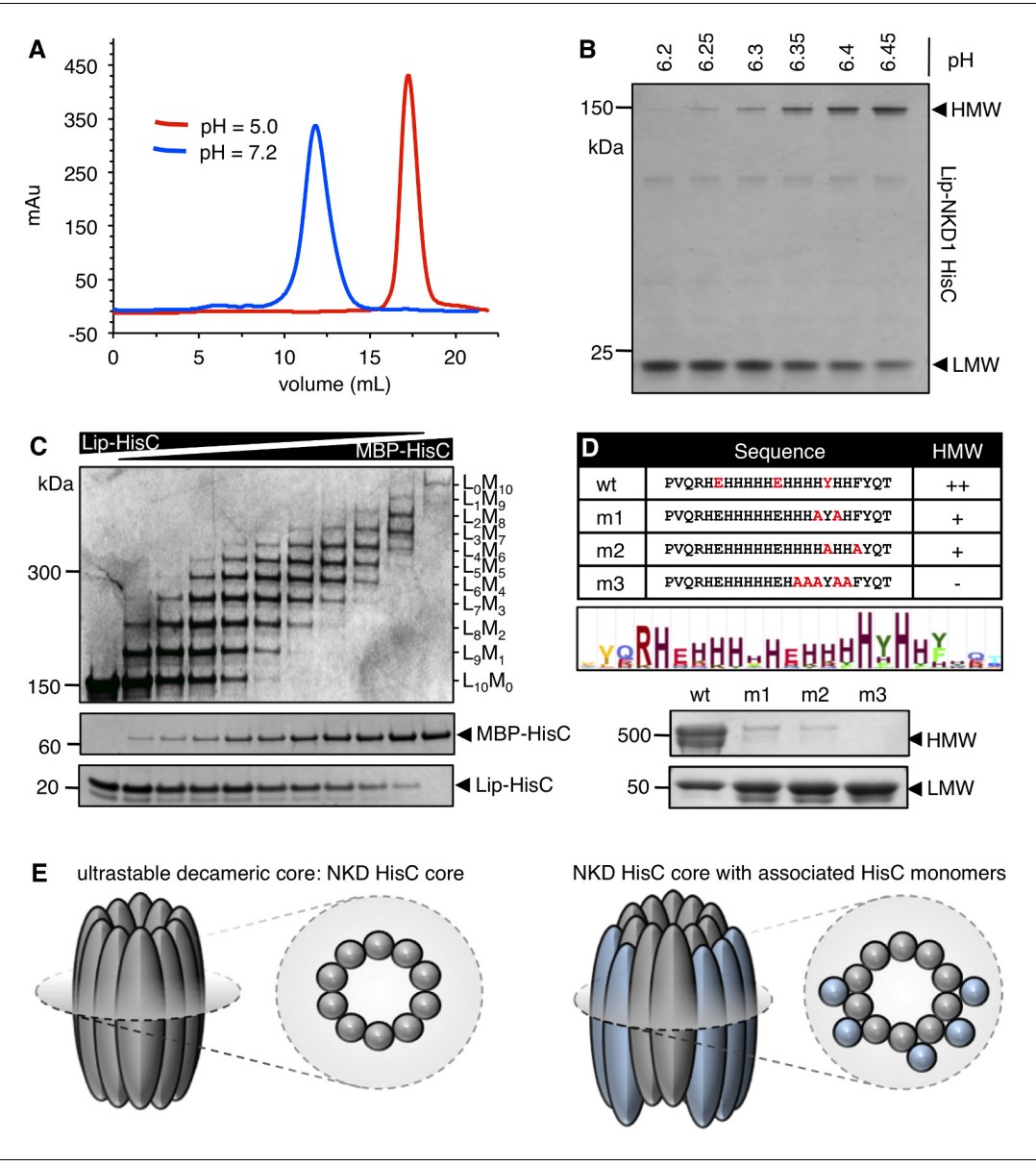

**Figure 2.** NKD1 HisC forms ultra-stable decameric core aggregates. (**A, B**) pH-dependent aggregation of purified recombinant Lip-NKD-HisC in vitro, monitored by (**A**) SEC or (**B**) PAGE. (**C**) Co-aggregation assays after mixing Lip-NKD-HisC and MBP-NKD-HisC at different ratios, revealing a decameric core (note that the self-aggregation of MBP-NKD-HisC is considerably less efficient than that of Lip-NKD-HisC, presumably owing to the large size of the MBP tag). (**D**) Mutational analysis, revealing that in vitro aggregation depends on conserved aromatic and charged residues within HisC. (**E**) Cartoon of NKD-HisC core and associated HisC monomers.

The online version of this article includes the following figure supplement(s) for figure 2:

**Figure supplement 1.** Effects of overexpressed Nkd1 HisC mutants on β-catenin signaling.

aggregation observed in vitro is required for the activity of Nkd1 in antagonizing DVL2 signaling in cells.

His-rich sequences are relatively rare in the genome. We searched the human genome for proteins exhibiting a minimum of eight His residues within a 13 amino acid stretch, which identified 74 proteins (containing 78 HisC sequences; *Figure 3—source data 1*). Most of these are nuclear proteins and thus unlikely targets for Naked/NKD, which is anchored at the plasma membrane by its N-terminus via myristoylation (*Chan et al., 2007*; *Hu et al., 2010*). The non-nuclear set contains

Axin1 and Prickle3 (a non-canonical Wnt signaling component) as well as a small number of additional proteins (*Figure 3A*). We tested a subset of these HisC (tagged with MBP) for in vitro co-aggregation, and found that MBP-Axin1-HisC exhibits the most pronounced co-aggregation with Lip-NKD-HisC (<5 molecules per core, at equivalent ratio; *Figure 3B*). Axin2 also exhibits an internal HisC, which co-aggregates with Lip-NKD-HisC (<4 molecules per core, at equivalent ratio), although this seems insufficient for its co-aggregation with ultra-stable Nkd1 HisC cores in cells (*Figure 3—figure supplement 1*). The MBP-HisCs from the remaining tested HisC-containing proteins co-aggregate poorly, or not at all, with Lip-NKD-HisC (*Figure 3—figure supplement 2*). Given the sequence conservation of the HisC in Axin orthologs, the specificity and avidity of co-aggregation between Axin1-HisC and NKD1 HisC cores in vitro and the co-aggregation between Axin1 and Nkd1 in cells, it seems likely that Axin1 is the primary if not only physiological target for NKD co-aggregation within the human genome.

| A | HisC sequence | HMW | Co-aggregate |
|---|---|---|---|
| NKD1 | HEHHHHHEHHHHYHH | +++ | +++ |
| Axin1 | HHHRHVHHHVHH | - | ++ |
| Prickle3 | HHHHNHHHHHNRH | - | - |
| USP34 | HHHHHHHHHHHHDGH | - | -/+ |
| CBL | HHHHHHHLSPH | - | - |
| DLGAP3 | HTSHHHHHHHHHHHHQSRH | - | + |
| FoxF2 | HAHPHHHHHHHVPH | - | - |
| CDX2 | HPHHHPHHHPHH | - | - |
| NLK | HHHHHHHHLPHLPPPHLHHH HHPQHHLH | - | - |
| SIAH3 | HPHHLSHHHCHHRHHHHLR HHAHPHHLHH | - | - |
| EPB41L4B | HHHQHQHQHQHQHH | - | - |
| ZIC3 | HHHHHHHHHHH | - | - |

**Figure 3.** NKD1 HisC co-aggregates with Axin1 HisC. (**A**) Selected HisC-containing proteins in the human genome (see also text) and their ability to co-aggregate with NKD1 HisC. (**B**) Co-aggregation assays after mixing Lip-NKD-HisC and MBP-Axin1-HisC at different ratios (MWM, molecular weight markers).

The online version of this article includes the following source data and figure supplement(s) for figure 3:

**Source data 1.** HisC proteins in the human genome and their subcellular location.
**Figure supplement 1.** Co-aggregation between NKD1 and Axin2.
**Figure supplement 2.** Systematic co-aggregation tests with NKD1-HisC.

## Dishevelled promotes co-aggregation between Nkd1 and Axin1

The level of SDS-resistant HMW Nkd1 aggregates clearly depends on the level of Nkd1 overexpression, indicating that HisC aggregation is concentration-dependent (*Figure 4A*; note that this concentration-dependence of HisC aggregation explains why the ratios between HMW and LMW Nkd1 vary somewhat between experiments, depending on the transfection efficiency). We therefore asked whether Wnt stimulation (which increases Nkd1 expression; *Van Raay et al., 2007*; *Yan et al., 2001*) would promote Nkd1 aggregation, which is indeed the case (*Figure 4B*). In addition, Dishevelled stimulates efficient Nkd1 aggregation since the formation of HMW aggregates is much reduced in DVL null cells (lacking all three DVL paralog; *Gammons et al., 2016b*; *Figure 4C*).

To determine whether DIX-dependent polymerization or DEP-dependent dimerization of Dishevelled is required for promoting HisC-dependent aggregation of Nkd1, we tested a panel of DVL2-GFP mutants (*Gammons et al., 2016a*) for their ability to restore formation of Nkd1 HMW aggregates upon re-expression in DVL null cells. We thus found that polymerization-deficient mutants or DVL2 lacking its DIX domain were most active, while DVL2 mutants whose DEP-dependent dimerization is blocked (E499G or G436P) were least active (*Figure 4D*; *Figure 4—figure supplement 1*). This indicates that the formation of stable DVL2 dimers formed by DEP domain swapping is crucial for Nkd1 aggregation. As expected, restoration of HMW aggregates also depends on the PDZ domain, the NKD-binding domain of DVL2 (*Rousset et al., 2001*; *Wharton et al., 2001*; *Yan et al., 2001*), but is hardly affected by K446M, which blocks binding of the DEP domain to the Frizzled receptor (*Figure 4D*).

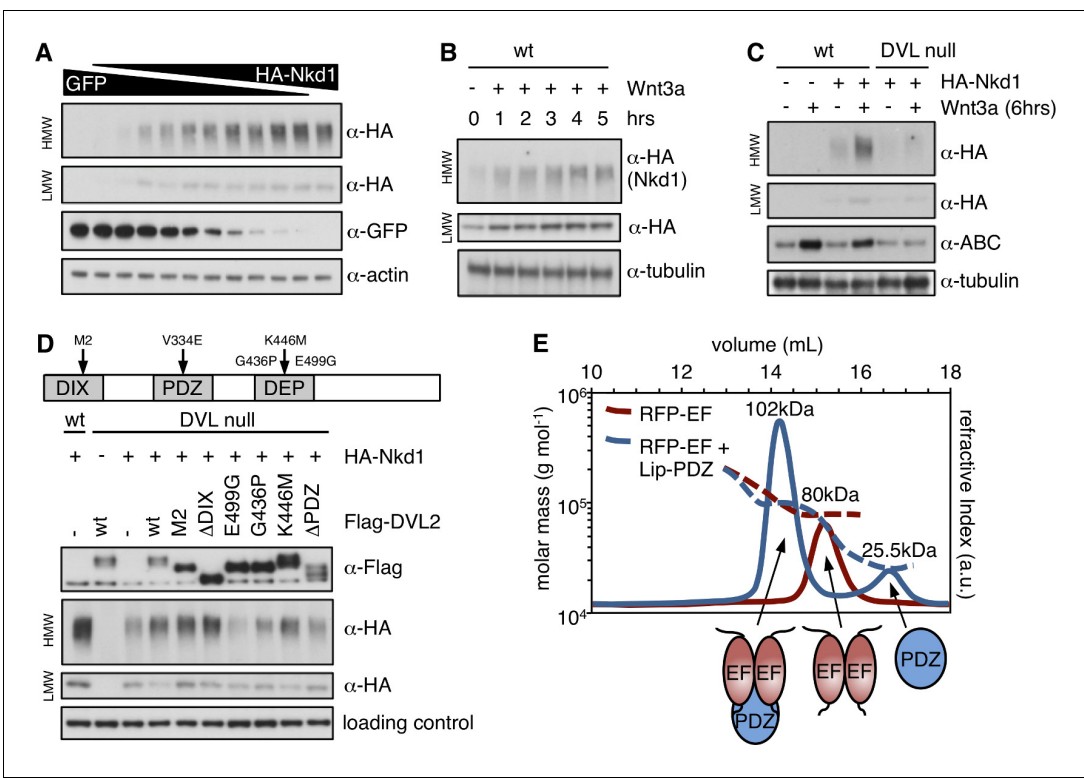

**Figure 4.** HisC aggregation is promoted by DVL2. (**A–D**) Western blots monitoring the formation of HisC aggregates in wt or mutant HEK293T cells dependent on (**A**) Nkd1 concentration, (**B**) duration of Wnt stimulation, (**C**) DVL or (**D**) complementation of DVL null cells by re-expression of wt or mutant DVL2 (E499G, G436P block DVL2 dimerization; K446M blocks Frizzled receptor binding) (*Gammons et al., 2016a*). (**E**) SEC-MALS, monitoring complex formation between purified recombinant RFP-NKD1-EF and Lip-DVL2-PDZ (as indicated by cartoons); numbers correspond to Mr values determined by MALS (expected: RFP-NKD1-EF dimer, 71.8 kDa; Lip-DVL2-PDZ, 22.0 kDa; complex, 93.8 kDa).

The online version of this article includes the following figure supplement(s) for figure 4:

**Figure supplement 1.** Binding of the NKD1 EF-hand to the DVL2 PDZ cleft.

To confirm that the DVL2 PDZ domain binds directly to the Nkd1 EF-hand, we used SEC followed by multi-angle light scattering (SEC-MALS) of purified recombinant domains to show that a stable complex forms between them (*Figure 4E*). Using isothermal titration calorimetry (ITC), we estimated a binding affinity in the low-micromolar range ($K_d 3.2 \pm 0.2$ μM), which depends on the integrity of the PDZ cleft as demonstrated by the failure of the EF-hand to bind to the V334E cleft mutant (*Figure 4—figure supplement 1*) whose binding to cognate substrate motifs is abolished (*Gammons et al., 2016b*). As expected, this cleft mutant also fails to restore Nkd1 HMW aggregates in DVL null cells (*Figure 4—figure supplement 1*). Interestingly, the molecular mass of the EF-hand-PDZ complex indicates that the PDZ domain binds to an EF-hand dimer (*Figure 4E*). Taken together with our previous results (*Figure 4D*), this implies that a single DVL2 dimer can bind to four molecules of Nkd1, resulting in a considerable increase of its local concentration, which may provide the trigger for the formation of HisC core aggregates.

## NKD HisC is required for Wnt-induced Axin destabilization

Our in vitro and cell-based assays uncovered the striking activity of the NKD HisC to form hyperstable aggregates and co-aggregates with Axin. However, these assays are based on recombinant NKD HisC or on overexpression of Nkd1 in cells. We therefore decided to use CRISPR to engineer specific HisC deletions in both NKD paralogs in HEK293T cells (NKDΔHisC), to test the physiological relevance of these histidine clusters and their interaction with Axin. As a comparison, we also generated double-knockout cells (NKD null cells), to assess the function of NKD on Wnt signaling and Axin stability (*Figure 5—figure supplement 1*).

Somewhat to our surprise, NKD null cells show a normal response to Wnt3a during the first ~ 3 hr (hrs) (*Figure 5A,B*), with no detectable hyperactivation of β-catenin-dependent transcription. However, by ~ 6 hr of Wnt3a stimulation, their signaling response is noticeably attenuated and begins to plateau, unlike the response of parental control cells which continues to rise, reaching at least three times the levels compared to that of the NKD null cells by 12 hr (*Figure 5A*). This is observed in two independently isolated NKD null lines, based on different targeting events (*Figure 5—figure supplement 1*). Similarly, the Wnt response is also significantly reduced from 6 hr onwards in two independently isolated NKDΔHisC lines, although this reduction is less pronounced than in NKD null cells (*Figure 5A*). Similar trends are observable if the levels of endogenous activated β-catenin are monitored with an antibody specific for the activated form (α-ABC; *Figure 5B*; *Figure 5—figure supplement 2*). These results provide clear evidence that NKD functions as a positive regulator of Wnt signaling in these human epithelial cells, and reveal the functional importance of its histidine cluster in the maintenance of Wnt responses during prolonged Wnt stimulation.

The reduction of β-catenin signaling in NKD-deficient cells coincides roughly with the half-time (t1/2) of the known Wnt-induced destabilization of Axin1 (*Ji et al., 2017*; *Yamamoto et al., 1999*). Indeed, while the levels of endogenous Axin1 gradually decrease to approximately half within the first 6 hr after Wnt stimulation in parental HEK293T cells, they decrease less in NKDΔHisC cells (*Figure 5B*), and there is no statistically significant reduction of the Axin1 levels in NKD null cells (*Figure 5—figure supplement 2*). Similarly, the levels of endogenous DVL2 are elevated in NKDΔHisC (*Figure 5B*) and NKD null cells (*Figure 5—figure supplement 2*), most noticeable for the Wnt-induced phosphorylated form. Thus, Axin1 and DVL2 are both stabilized during prolonged Wnt stimulation in cells without functional NKD, indicating that NKD functions to target both signalosome components for degradation in these cells. Its requirement for maintaining β-catenin responses can readily be explained by its destabilizing effect on Axin1 but not on DVL2 (see also Discussion), which argues that Axin1 rather than DVL2 is rate-limiting for a sustained Wnt response of these cells.

To test this further, we asked whether Nkd1 might accelerate the Wnt-induced degradation of Axin1 and DVL2 upon overexpression. Indeed, overexpressed Nkd1 but not Nkd1ΔHisC (incapable of forming HMW aggregates) accelerates the destabilization of endogenous Axin1 upon Wnt stimulation, but does not affect the levels of endogenous DVL2 (*Figure 5C*). This provides further support for the critical role of the Nkd1 HisC in the Wnt-induced destabilization of Axin1, and suggests that Axin1 rather than DVL2 is the relevant target of NKD in persistently Wnt-stimulated HEK293T cells.

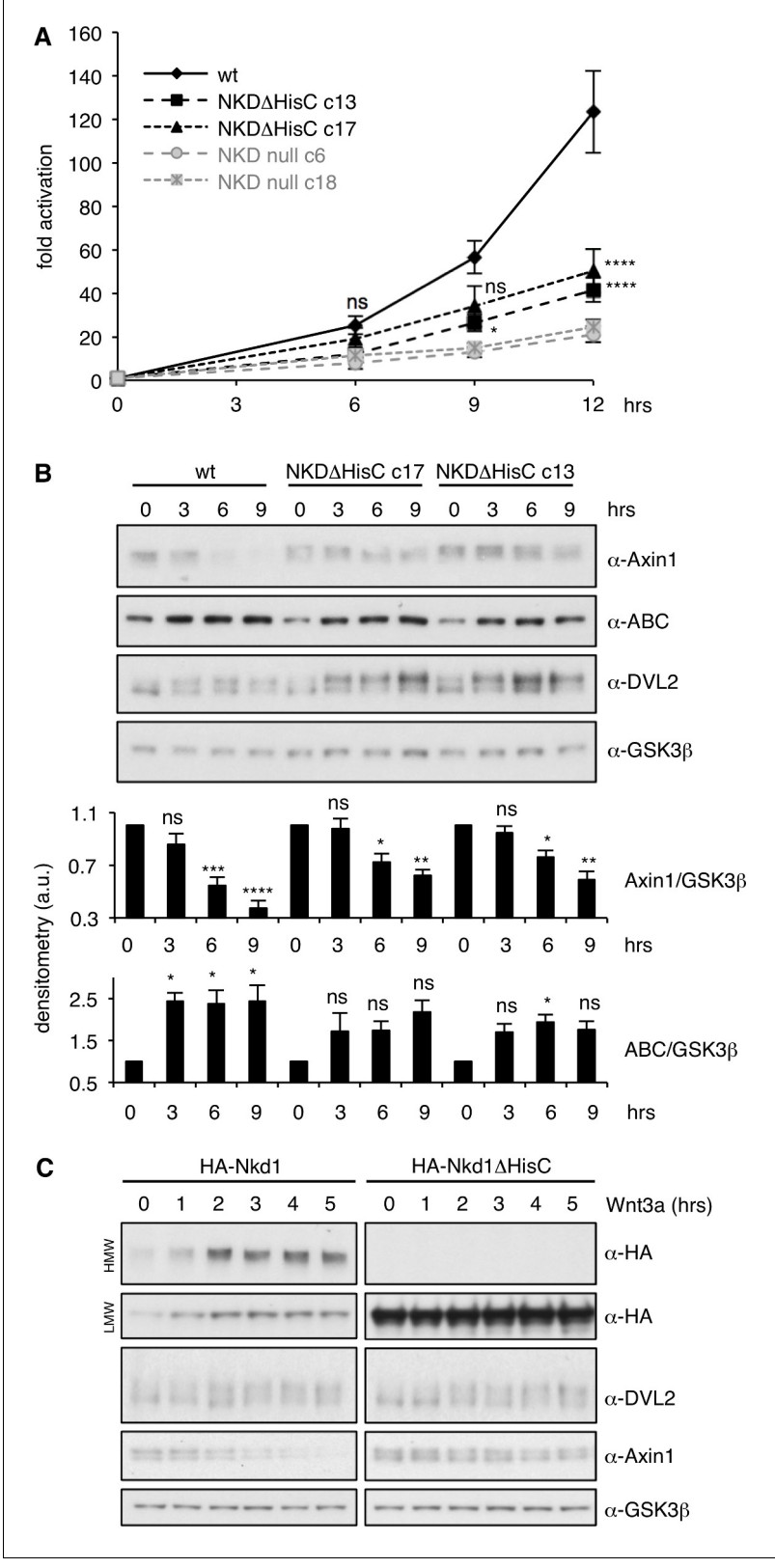

**Figure 5.** NKD is required for sustained Wnt signaling. (**A**) SuperTOP assays in HEK293T cells, monitoring the signaling response of NKDΔHisC, NKD null-mutant or control cells after 6 hr of Wnt3a stimulation (two independently isolated lines each, see also *Figure 5—figure supplement 1*); error bars, SEM of > 3 independent experiments; two-way ANOVA with multiple comparisons (for c13 and c17; see *Figure 5—figure supplement 2*

*Figure 5 continued on next page*

*Figure 5 continued*

for null lines); *p < 0.05, ****p < 0.0001. (**B**) Representative western blots monitoring expression levels of endogenous proteins in two independent NKDΔHisC lines following 0–9 hr of Wnt stimulation (as indicated), quantified by densitometry of >3 independent experiments relative to GSK3β as internal control (a.u., arbitrary units); error bars, SEM of > 3 independent experiments; two-way ANOVA with multiple comparisons; *p < 0.05, **p < 0.01,***p < 0.001, ****p < 0.0001, ns = not significant. For the corresponding analysis of NKD null cell lines, see *Figure 5—figure supplement 2*. (**C**) Western blots monitoring the effects of overexpressed wt or ΔHisC HA-Nkd1 on the levels of endogenous proteins (indicated on the right; GSK3β, internal control) after 0–5 hr of Wnt stimulation.

The online version of this article includes the following source data and figure supplement(s) for figure 5:

**Source data 1.** Oligonucleotides used for CRISPR engineering of HEK293T cells and confirmation of lesions.
**Figure supplement 1.** CRISPR engineering of HEK293T cells.
**Figure supplement 2.** NKD is required for sustained Wnt signaling.

## The phenotype of *Drosophila nkd*$^{\Delta his}$ mutants implies attenuated Wingless signaling

Given the unexpected reduction of β-catenin signaling in NKD-deficient human cells, we decided to revisit the *nkd* mutant phenotype in *Drosophila* embryos whose signature 'naked cuticle' in a putative null allele (*nkd*$^{7H16}$) indicates hyperactive Wingless signaling (*Zeng et al., 2000*; *Figure 6A,B*). However, these authors noted a caveat regarding a possible genetic interaction with *h*$^1$, a recessive marker born on the original chromosome (*Nusslein-Volhard et al., 1984*). Indeed, the cuticle phenotype of another strong allele (*nkd*$^{7E89}$) is far milder upon removal of the *h*$^1$ allele (*Waldrop et al., 2006*; *Figure 6C,D*).

To determine the true *nkd* null phenotype, we generated a CRISPR-engineered null allele (*nkd*$^{13}$) truncating Naked at codon 172 upstream of its EF-hand (*Figure 6—figure supplement 1*). Freshly hatched *Drosophila* larvae show eight ventral abdominal denticle belts comprising six denticle rows each (*Figure 6A*), whereby the denticles in each row exhibit a characteristic shape and orientation, specified by row-specific selector gene products (e.g. Engrailed, En) and by positional signals such as Wingless (Wg), Hedgehog and Spitz (an Epidermal Growth Factor-like factor activated by the transmembrane protease Rhomboid) (*Hatini and DiNardo, 2001*; *Martinez Arias et al., 1988*; *O'Keefe et al., 1997*; *Szüts et al., 1997*; *Figure 7A*). As expected, homozygous *nkd*$^{13}$ embryos exhibit stretches of excess naked cuticle, however, these mutant embryos also show near-normal denticle belts that are uninterrupted by naked patches (*Figure 6E*), more so than the mutant embryos bearing *h*$^1$ (*Figure 6B,C*). Notably, patches of missing denticles in *nkd*$^{13}$ embryos are predominantly observed in rows 3–7 (*Figure 7B*), and seem to correlate with the sporadic patches of ectopic Wg expression (*Martinez Arias et al., 1988*; *Figure 7—figure supplement 1*), owing to a failure of En and its *rhomboid* target gene to repress *wg* posteriorly to their own expression domains (*Zeng et al., 2000*; *O'Keefe et al., 1997*; *Szüts et al., 1997*; *Figure 7A*).

On close examination of the near-normal denticle belts in *nkd*$^{13}$ embryos, we noticed ectopic small denticles anteriorly to these belts (*Figure 7C*), i.e. in row 0 (*Figure 7A*). This phenotype, observed in ~ 20% of near-normal denticle belts of *nkd*$^{13}$ homozygotes (*Figure 7C*), signifies a reduced Wg response in this part of the segment. This is not a CRISPR off-target effect since we also observe excess row-0 denticles in various transheterozygous embryos, including transheterozygotes of *nkd*$^{13}$ and *h*$^1$ *nkd*$^{7H16}$ and, occasionally, in the rare near-normal denticle belts of *h*$^1$ *nkd*$^{7H16}$ homozygotes (*Figure 7—figure supplement 2*). These excess row-0 denticles are consistent with the phenotype in NKD null-mutant human cells (*Figure 5A*), and likely to reflect a failure to maintain Wg responses through late embryonic stages (a condition that causes excess small denticles since reduced Wg signaling is permissive for denticle specification by Rhomboid and its Spitz substrate, which stimulates EGF receptor signaling; *O'Keefe et al., 1997*; *Szüts et al., 1997*). We also examined the expression of a Wg-responsive reporter from the *Ultrabithorax* gene (*UbxB.lacZ*) (*Thüringer et al., 1993*) as a direct readout of Wg signaling in the middle midgut of late-stage embryos (*Riese et al., 1997*), and found this to be reduced consistently in *nkd*$^{13}$ mutant embryos (*Figure 7—figure supplement 3*). This further underscores the notion that Naked functions in late embryonic stages to maintain Wg signaling responses.

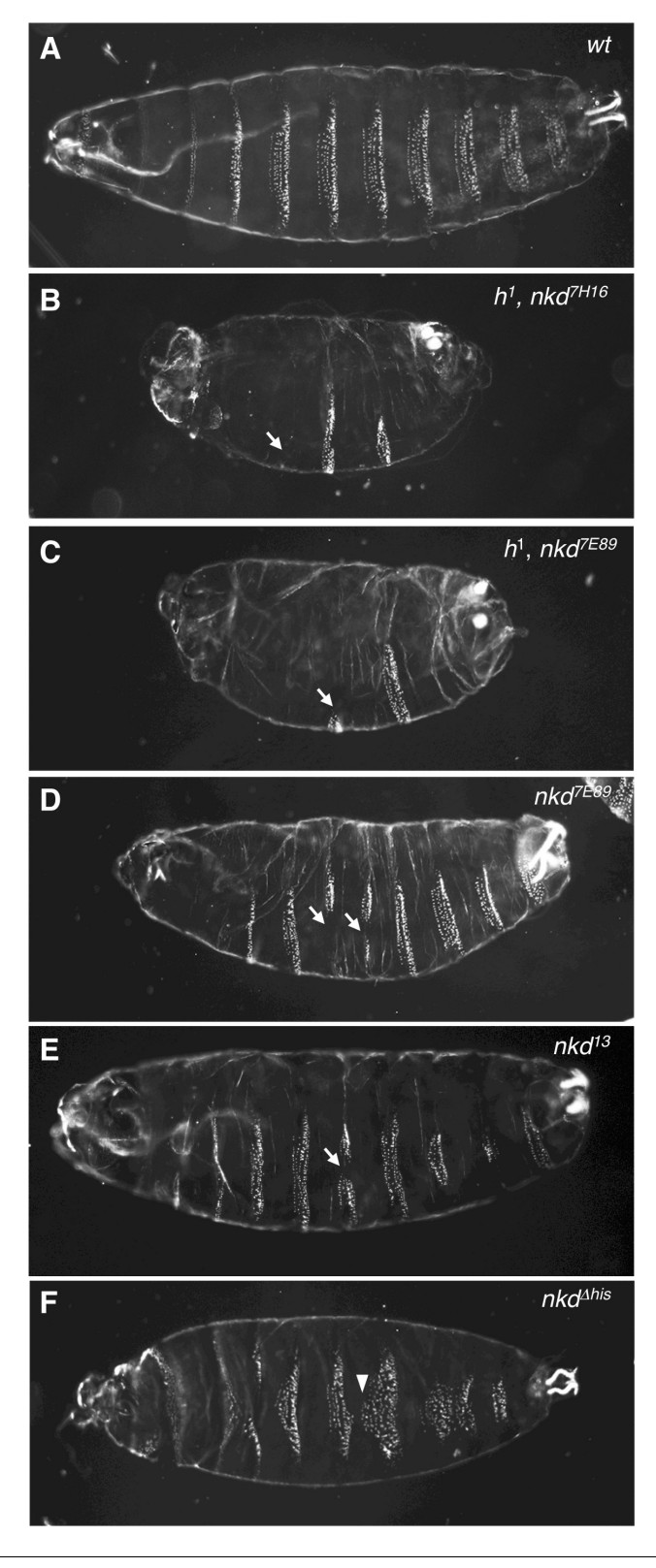

**Figure 6.** Allele-dependent Wg signaling defects in *nkd* mutant *Drosophila* embryos. Dark-field images of ventral cuticles of (**A**) wt and (**B–F**) homozygous mutant embryos, as indicated in panels (*nkd*$^{7H16}$, *nkd*$^{7E98}$, pre-existing alleles; *nkd*$^{13}$, *nkd*$^{Δhis}$ CRISPR-engineered alleles), showing regions of excess naked cuticle (arrows) signifying ectopic Wg signaling. Note that the 'near-naked' phenotype is only seen in embryos bearing *h*$^{1}$ (**B, C**); excess

*Figure 6 continued*

naked cuticle regions are seen infrequently in null-mutant embryos without $h^1$ (D, E), or not at all in $nkd^{\Delta his}$ embryos bearing a HisC deletion (F) whose most penetrant phenotype are excess small denticles (arrowheads) signifying reduced Wg signaling.

The online version of this article includes the following source data and figure supplement(s) for figure 6:

**Source data 1.** Oligonucleotides used for Drosophila CRISPR engineering and confirmation of lesions.
**Figure supplement 1.** CRISPR-engineered *Drosophila nkd* alleles.

---

Given our results from human cells implicating the NKD HisC in maintaining prolonged Wnt responses (*Figure 5A*), we asked whether this might also be true for the HisC of *Drosophila* Naked. We therefore generated a deletion of the C-terminal HisC ($nkd^{\Delta his}$) by CRISPR engineering (*Figure 6—figure supplement 1*). Homozygous $nkd^{\Delta his}$ flies can be obtained, however ~ 20% of their offspring die as late-stage embryos. Each of these embryos shows several abnormal denticle belts with disordered denticle rows, misoriented denticles as well as excess small denticles in their anterior regions (*Figure 6F* and *7C*), most likely including excess row-0 denticles since these can also be observed in transheterozygous $nkd^{\Delta his}/nkd^{13}$ embryos (*Figure 7—figure supplement 2*). It thus appears that these excess anterior denticles in $nkd^{\Delta his}$ mutant embryos parallels the Wnt signaling defect of the NKDΔHis human cells (*Figure 5A*), implicating the C-terminal HisC of Drosophila Naked in the maintenance of the Wg response throughout late embryonic stages.

## Nkd1 HisC aggregates associate with the p62 autophagy receptor

To identify potential effectors of Naked/NKD in maintaining prolonged Wnt responses, we used a BioID proximity-labeling approach, tagging Nkd1 with the promiscuous biotin ligase BirA* (*Roux et al., 2012*) internally (within its non-conserved low-complexity linker sequences), to avoid interfering with its N-terminal myristoylation (*Hu et al., 2010*) or with the function of its C-terminal HisC. We confirmed that Nkd1-BirA* is as active as HA-Nkd1 in suppressing DVL2 signaling (J. E. F., thesis). We then introduced this bait as an inducible transgene into a specific genomic location of Flp-In-T-REx-293 cells to keep its expression low, as previously described (*van Tienen et al., 2017*). As controls, we used mutant Nkd1-BirA* versions bearing a point mutation in their N-terminal myristoylation site (G2A), or a deletion of their HisC (ΔHisC; *Figure 8A*). Amongst the hits from this approach, we expected to identify DVL and Axin paralogs as direct Nkd1-binding proteins, but also bystanders or 'vicinal' proteins (*Roux et al., 2012*) that bind to, or are closely associated with, DVL or Axin. We considered hits as specific only if the spectral counts obtained with wt Nkd1-BirA* were > 5 fold higher than those obtained with GFP-BirA*.

Cells expressing BirA* baits were stimulated with Wnt3a for 12 hr while also being incubated with biotin, to label bait-interacting proteins. As expected, we found DVL1-3 amongst our top hits (*Figure 8B*; *Figure 8—source data 1*). We also found components of the β-catenin destruction complex, namely Axin1 itself and its key binding partner, the APC tumour suppressor, as well as AMER1 and β-catenin, suggesting that the entire complex is associated with Nkd1 HisC aggregates in Wnt-stimulated cells. Also on our list were the Wnt co-receptors LRP5 and LRP6 whose cytoplasmic tails form stable complexes with Axin upon Dishevelled-dependent phosphorylation (*Bilic et al., 2007*; *Tamai et al., 2004*). Our myristoylation-dependent hits include the non-canonical Wnt signaling components VANGL1, VANGL2, CELSR1, and ROR2, as well as Notch1 and Notch2 (physiological substrates of Dishevelled-activated NEDD4 family ligases) (*Mund et al., 2015*; *Figure 8B*; *Figure 8—source data 1*). Strikingly, our top HisC-dependent hits are the two main autophagy receptors p62 and NBR1 (*Kirkin et al., 2009*). We used BioID proximity-labeling (BioIP) to confirm that Nkd1-BirA* binds to endogenous p62 in a HisC-dependent fashion (*Figure 8C*).

Dishevelled is a known substrate for selective autophagy, and is targeted to autophagosomes via association with p62, LC3 and GABARAPL and, ultimately, to lysosomes for degradation (*Gao et al., 2010*; *Ma et al., 2015*; *Zhang et al., 2011b*). However, the association between Nkd1 and p62 is independent of Dishevelled since the two proteins coIP as efficiently in DVL null cells as in control HEK293T cells (*Figure 8D*). We used internal deletions to map the p62-interacting domain of Nkd1 to its EF-hand (*Figure 8D*), which indicates that this EF-hand can bind Dishevelled and/or p62. As expected from our BioID screen, co-expressed Flag-p62 shows a strong preference for binding to

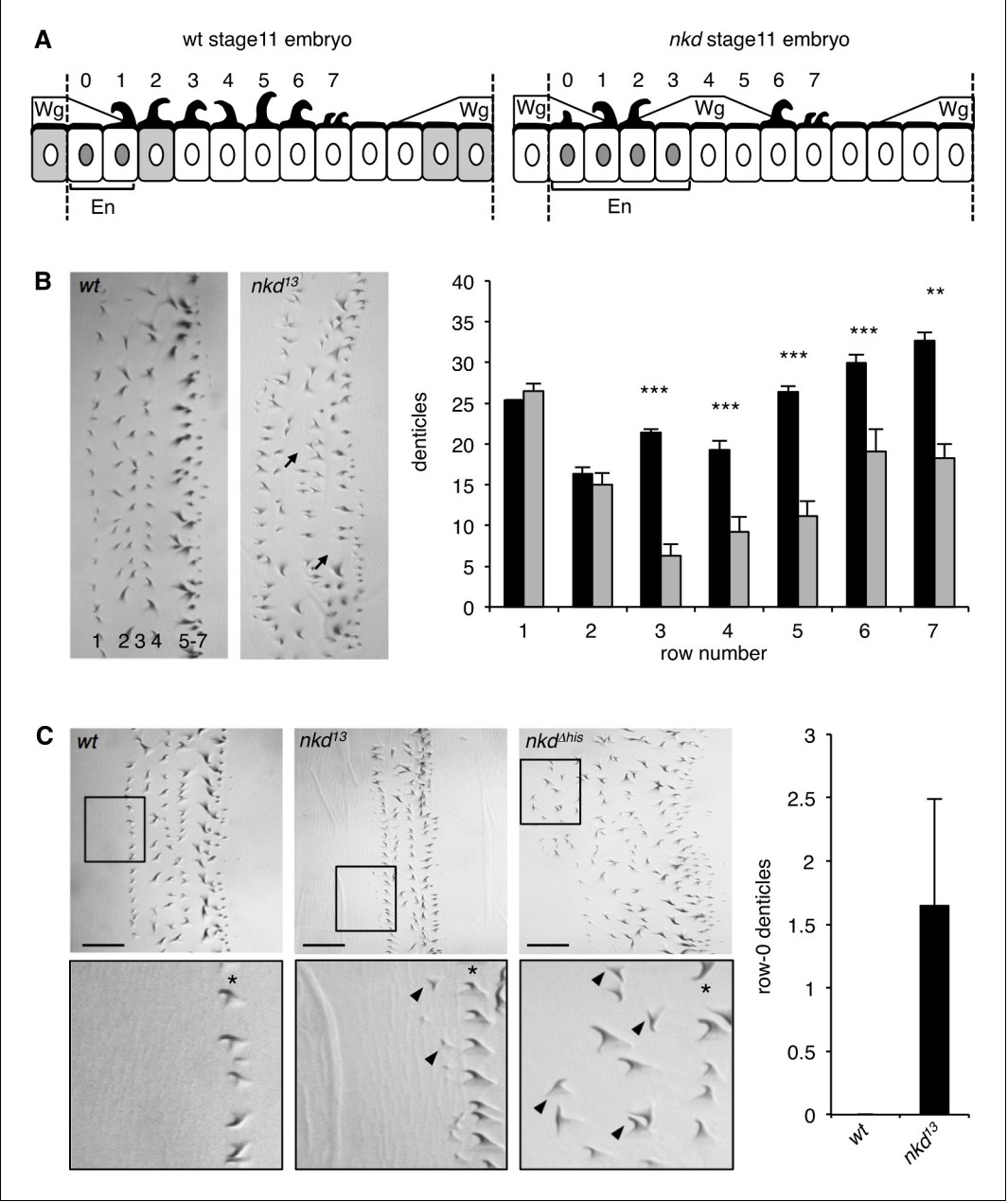

**Figure 7.** Quantification of context-dependent Wg signaling defects in embryos bearing *CRISPR alleles*. (**A**) Cartoons of row-specific ventral denticles relative to expression domains of Wg, En, and Nkd in stage 11 wt (left) and *nkd¹³* null-mutant (right) embryos; numbers indicate denticle rows; see also text. (**B**) *Left*, examples of abdominal denticle belts in wt and *nkd¹³* homozygous embryos, with arrows pointing to missing denticles; *right*, quantitation of numbers of denticles per row in wt (black bars) or *nkd¹³* null-mutant (white bars) embryos (n = 10 near-normal denticle belts); Student's unpaired t-test; **p < 0.01, ***p < 0.001. (**C**) *Left*, *nkd¹³* and *nkdᐞʰⁱˢ* homozygous but not wt embryos exhibit excess row-0 denticles (arrowheads; asterisks mark row-1); *right*, quantitation of excess row-0 denticles in *nkd¹³* mutants (n = 20 near-normal denticle belts).

The online version of this article includes the following figure supplement(s) for figure 7:

**Figure supplement 1.** Ectopic Engrailed and Wingless expression in *nkd* mutant *Drosophila* embryos.

**Figure supplement 2.** Excess anterior denticles in transheterozygous embryos bearing different combinations of *nkd* alleles.

**Figure supplement 3.** Reduced expression of Wg-responsive *Ubx* midgut reporter gene in *nkd* null-mutant embryos.

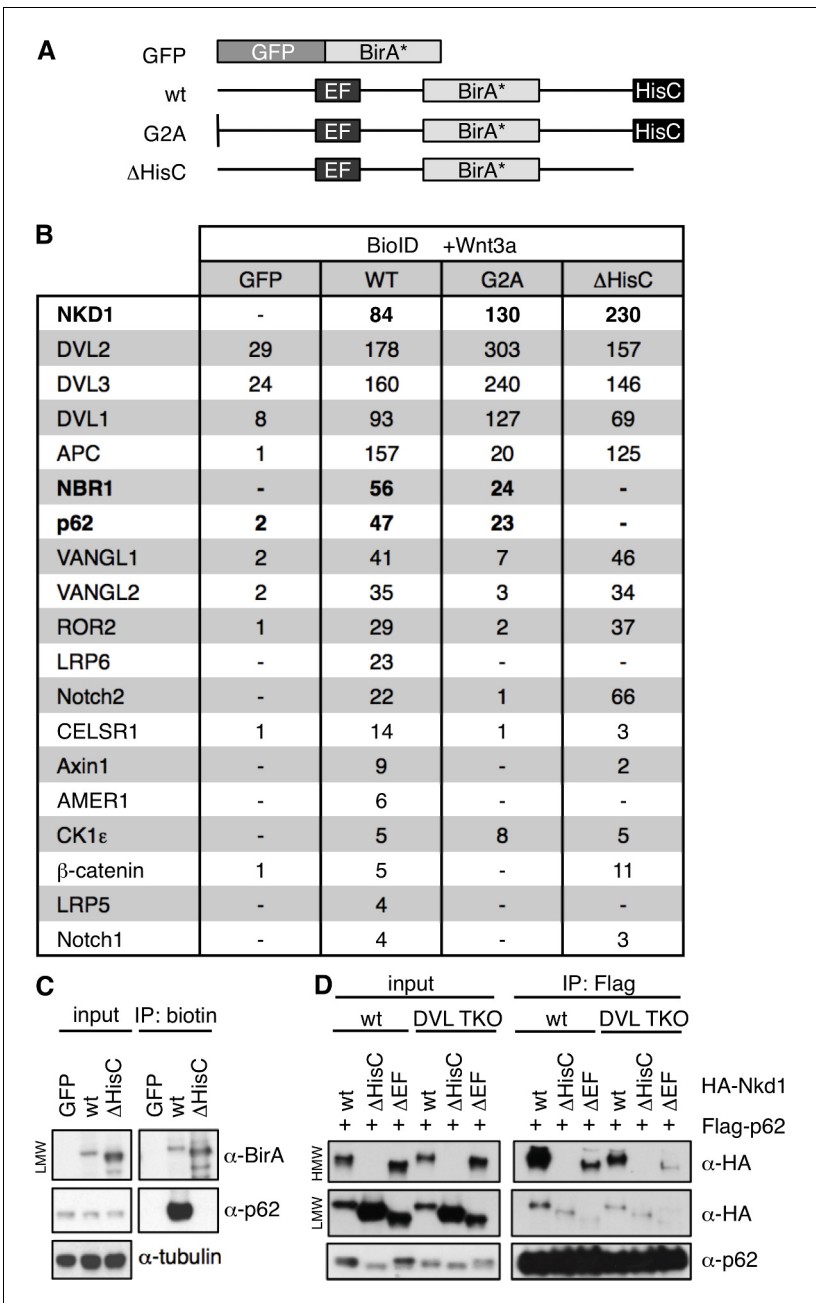

**Figure 8.** p62 is a HisC-dependent interactor of Nkd1. (**A**) Cartoons of BirA* baits used for BioID proximity labeling. (**B**) Selected BioID hits obtained with wt or mutant Nkd1-BirA* stably integrated in HEK293 cells (see also text); numbers correspond to unweighted spectral counts > 95% probability (for full list, see *Figure 8—source data 1*); note that APC is large (~311 kDa) and relatively abundant, explaining the high numbers of spectral counts. (**C**) BioID proximity-labeling assays in HEK293 cells with stably integrated baits, monitoring association between endogenous p62 and Nkd1-BirA* or Nkd1ΔHisC-BirA* (only LMW Nkd1-BirA* is shown since HMW Nkd1-BirA* is too large to be detectable on blots). (**D**) CoIP assays between proteins co-expressed in HEK293T cells, as indicated above panels, monitoring association between HA-p62 and wt or mutant HA-Nkd1.

The online version of this article includes the following source data for figure 8:

**Source data 1.** Full list of BioID hits obtained with wt or mutant Nkd1-BirA*.

HMW HA-Nkd1 (*Figure 8D*). These results suggest that p62 may bind to Nkd1 HisC aggregates and Nkd1-Axin1 co-aggregates, to target these for autophagy-dependent degradation (see Discussion).

## Discussion

From numerous studies in *Drosophila* and vertebrates, a picture has emerged of Naked/NKD as a widespread Wnt-inducible signaling component that downregulates canonical and non-canonical Wnt responses by binding to and destabilizing Dishevelled (see Introduction). Our results from CRISPR-engineered null-mutants in Drosophila and human epithelial cells lead to a revised picture which also includes positive regulatory effects of Naked/NKD on canonical Wnt signaling, likely mediated by its binding to and destabilizing Axin during prolonged Wnt stimulation. Therefore, whether Naked/NKD antagonizes or promotes Wnt signaling appears to depend on whether Dishevelled or Axin is limiting for Wnt responses in any given cell. This context-dependence of Naked/NKD renders it a versatile feedback regulator of the pathway, and may also explain why it is dispensable in some cells. Feedback control has long been recognized as a widespread mechanistic principle in cellular signaling that leads to canalization and robustness of development (*Freeman, 2000*).

Both negative and positive regulatory effects of Naked/NKD on Wnt signaling are mediated by its activity to destabilize its interaction partners. We discovered a rather unusual molecular mechanism underlying this destabilization, which depends on ultra-stable aggregation of Naked/NKD via a highly conserved cluster of histidines in the Naked/NKD C-terminus. NKD-HisC core aggregates are decameric, and their finite size implies a well-defined structure. Consistent with this, they co-aggregate specifically with a similarly conserved HisC in Axin1 – as far as we can tell, the only HisC-containing substrate in the human genome capable of co-aggregating with NKD efficiently, implicating Axin1 as its main if not only physiological target. Because of their oligomeric nature, Naked/NKD-HisC aggregates attain a high avidity for Axin-HisC, enabling these two proteins to interact efficiently.

Our preliminary searches have indicated that histidine clusters in NKD and Axin proteins are completely conserved throughout vertebrates, however their occurrence amongst invertebrates are more patchy. Indeed, some invertebrate species conta.in HisCs in both their NKD and Axin orthologs (e.g. Platynereis, Nematostella, and several gastropod and moskito species), some only in their NKD but not in their Axin orthologs (e.g. *Drosophila* and Apis), while others (e.g. nematodes) have HisCs in neither proteins (*Figure 1—figure supplement 1*). It is therefore possible that some invertebrate NKD proteins (i) use a different mechanism to co-cluster with Axin, (ii) do not form clusters to associate with Axin, or (iii) do not associate with Axin at all. It will be interesting to explore this in future studies.

The physiological relevance of the HisC-dependent aggregation of Naked/NKD is indicated by specific HisC deletions in human cells: their signaling defects following prolonged Wnt stimulation (reduced β-catenin responses, and stabilization of Axin; *Figure 5A,B*; *Figure 5—figure supplement 2*) are consistent with precocious termination of Wnt signaling by Axin which, as a result of its stabilization, resumes assembling active β-catenin destruction complexes prematurely. It therefore seems that Naked/NKD acts similarly to the ubiquitin E3 ligase SIAH which targets Axin for proteasomal degradation upon Wnt signaling (*Ji et al., 2017*). Indeed, the two factors may function redundantly to destabilize Axin, thereby maintaining Wnt signaling responses by preventing Axin from re-establishing quiescence too early.

The marked concentration-dependence of HisC core aggregation (*Figure 4A*) ensures that Naked/NKD only aggregates once it has accumulated upon prolonged Wnt signaling. Furthermore, aggregation is promoted by Dishevelled, and our evidence indicates that DEP-dependent dimerization of Dishevelled is a prerequisite for facilitating Naked/NKD aggregation. Since the PDZ domain of Dishevelled binds to a dimer of NKD EF-hands, this implies that the interaction between the two proteins results in a considerable increase (>4 fold) in local concentration of Naked/NKD, which may overcome the threshold necessary for triggering HisC aggregation. The dual dependence of Naked/NKD aggregation on Wnt stimulation and Dishevelled dimerization could safeguard against fortuitous spontaneous aggregation and, thus, inappropriate destabilization of Axin.

Given the moderately stable interaction between Dishevelled and Naked/NKD (*Figure 3—figure supplement 2*), it seems plausible that the two proteins remain associated following HisC aggregation, which could lead to the destabilization of Dishevelled alongside Axin. In support of this is the

stabilization of DVL2 in NKDΔHisC cells during prolonged Wnt stimulation (*Figure 5B*), which may explain why loss-of-function of Naked/NKD can lead to hyperactive Wnt signaling, and interfere with PCP signaling (*Hu et al., 2010*; *Marsden et al., 2018*; *Rousset et al., 2001*; *Schneider et al., 2010*; *Van Raay et al., 2007*; *Zeng et al., 2000*). Furthermore, it could also explain why overexpressed Naked/NKD downregulates the signaling activity of Dishevelled (*Rousset et al., 2001*; *Figure 1A*) and reduces its levels (*Guo et al., 2009*; *Hu et al., 2010*; *Schneider et al., 2010*). Indeed, it may explain why Naked/NKD loss has little effect on Wnt signaling in some context since its effects in destabilizing Dishevelled (a Wnt agonist) as well as Axin (a Wnt antagonist) simultaneously may cancel themselves out in some circumstances.

Our discovery of the p62 and NBR1 autophagy receptors as Nkd1-interacting proteins suggests that the destabilization of Naked/NKD-binding partners is mediated by autophagy. This notion is supported by the pronounced binding preference of p62 to aggregated Nkd1 (*Figure 8D*), presumably reflecting the high avidity of decameric NKD-HisC core aggregates for binding partners. This binding preference implies that the targeting of NKD-binding partners to autophagosomes is contingent on Naked/NKD aggregation. The notion of Naked/NKD targeting its binding partners for autophagy is also consistent with earlier reports that Dishevelled can be degraded by autophagy via binding to p62 (*Gao et al., 2010*; *Ma et al., 2015*; *Zhang et al., 2011b*). Indeed, there appears to be an interaction network between Dishevelled, p62, Axin and aggregated Naked/NKD, implicating Naked/NKD-HisC cores as adaptors between Wnt signalosomes and autophagy receptors, as an alternative route to the SIAH pathway (*Ji et al., 2017*) for the disposal of Axin and other signalosome components upon Wnt signaling. Since this route would involve the envelopment of Naked/NKD co-aggregates by autophagic membranes (*Kirkin et al., 2009*), it is worth noting that these co-aggregates may not inevitably reach lysosomes because of the endosomolytic properties of highly concentrated histidines (155–265 histidines per single Nkd1-Axin1 co-aggregate): these histidines are expected to become protonated under the mildly acidic conditions typical for endocytic compartments, which could cause swelling and rupture of autophagosomal membranes (*Lo and Wang, 2008*), thereby potentially allowing escape and recycling of Naked/NKD co-aggregates. This putative endosomolytic activity of Naked/NKD-HisC co-aggregates is intriguing, and worth testing in future under physiological conditions.

## Materials and methods

### Plasmid constructions

Mutagenesis of parental plasmid DNA was carried out with standard PCR-based methods, using either KOD DNA polymerase (Merck Millipore), Phusion DNA polymerase (NEB) or Q5 polymerase (NEB), and verified by sequencing. To generate BioID plasmids, the coding sequence for Nkd1 (and mutants thereof) and BirA*(R118G) were amplified by megaprimer PCR and inserted into pcDNA5/FRT/TO using Gibson assembly. The tag of HA-Nkd1 (*Rousset et al., 2001*) was switched to generate Flag-mNkd1.

### Cell-based assays

HEK293T and L cells were obtained from ATCC (authenticated by STR DNA profiling). Upon receipt, cells were frozen, and individual aliquots were taken into culture, typically for analysis within < 10 passages, and regularly tested for Mycoplasma infection. Cells were cultured in DMEM (GIBCO), supplemented with 10% fetal bovine serum (FBS) at 37°C in a humidified atmosphere with 5% $CO_2$. All cells were screened. In addition, cells used for CRISPR engineering were grown in media containing plasmocin (25 mg/ml) prior to DNA sequencing, to guard against Mycoplasma infection. Transient cell transfections were performed using polyethylenimine or Lipofectamine 2000. Wnt stimulation with Wnt3a-conditioned media was for 6 hr (unless otherwise stated).

For coIP assays, cells were lysed 24–36 hr post-transfection in lysis buffer (20 mM Tris-HCl pH 7.4, 200 mM NaCl, 10% glycerol, 0.2% Triton X-100, protease inhibitor cocktail and PhosSTOP). DNA amounts were adjusted for equal expression (1 µg HA-Nkd1; 200 ng HA-Nkd1ΔHisC; 300 ng HA-Nkd1ΔEF). Lysates were cleared by centrifugation for 10 min (min) at 16,100x *g*, and supernatants were incubated with affinity gel (Flag- or GFP-trap) for 90 min to overnight at 4°C. Subsequently,

immunoprecipitates were washed 4x in lysis buffer and eluted by boiling in LDS sample buffer for 10 min. Uncropped western blots are shown in *Supplementary file 1*.

For SuperTOP assays, HEK293T cells were lysed < 20 hr post-transfection with SuperTOP and CMV-Renilla (control) plasmids, and analyzed with the Dual-Glo Luciferase Reporter Assay kit (Promega) according to the manufacturer's protocol. Values were normalized to Renilla luciferase, and are shown as mean ± SEM relative to unstimulated controls (set to one in *Figure 1A* and *5A*; *Figure 1— figure supplement 2*; *Figure 5—figure supplement 2*).

## BioID *proximity labeling*

For BioID experiments, Flp-In HEK293 cell transfections were supplemented with pOG44 recombinase and selected with 250 μg ml$^{-1}$ hygromycin 48-hr post-transfection. Stable cell lines were induced with 1 μg ml$^{-1}$ tetracycline and treated with Wnt3a-conditioned media and 50 μM biotin 12 hr prior to lysis. BioID pull-downs were carried out using Streptavidin MyOne Dynabeads essentially as described (*Roux et al., 2013*), and biotinylated proteins eluted by boiling in LDS sample buffer. Samples were analyzed by SDS-PAGE (BioIP assays) or mass spectrometry (see below).

## Densitometry

Densitometry of western blots (*Figure 5A*; *Figure 5—figure supplement 2*) was performed on scanned low exposure films using ImageJ software. Where possible, an average of two exposures was taken for each experiment. Values are expressed as arbitrary units obtained for proteins of interest relative to GSK3β as loading controls. For each cell line, values were normalized to its own 0 hr control.

## CRISPR/Cas9 genome editing of human cells

Genome editing by CRISPR/Cas9 of HEK293T cells was essentially as described (*van Tienen et al., 2017*), using single-guide RNA-encoding plasmid derivatives of pSpCas9(BB)−2A-GFP (PX458; *Ran et al., 2013*). The CRISPR design tool crispor.tefor.net was used to design guide RNAs (*Figure 5—figure supplement 1*). Cells were selected for high expression of GFP by fluorescence-assisted cell sorting 48 hr post-transfection, and individual clones expanded in 96-well plates. Individual cell clones were screened by sequencing (*Figure 5—source data 1*) to confirm the presence of frameshifting lesions. To ensure consistency and to guard against off-target effects, multiple lines were isolated and characterized in each case.

## CRISPR/Cas9 genome editing in *Drosophila*

Single-guide RNA sequences (designed using crispor.tefor.net; *Figure 6—figure supplement 1*) were inserted into pCFD3 by *Bbs*I digestion, as previously described (*van Tienen et al., 2017*). Embryos from *nos.cas9* flies (from Simon Bullock) were collected for 1 hr and dechorionated in 8% sodium hypochlorite solution for 2 min before injection with DNA (200 ng μl$^{-1}$) under 10S oil. For genotyping, DNA was extracted from individual flies by standard methods, and sequences were determined after PCR amplification (*Figure 6—source data 1*).

## Analysis of embryonic cuticles

Embryos were collected overnight and staged for 24 hr at 25°C. Unhatched embryos were dechorionated in 8% sodium hypochlorite for 3 min, washed in water and vortexed for 1 min in 1:1 heptane: methanol (−20°C) to remove the vitelline membrane. Embryos from the bottom of the tube were transferred onto a slide and mounted in 1:1 lactic acid:Hoyer's-based medium (Hoyer's mountant), and slides were incubated at 65°C overnight. Imaging of cuticles was done with a Zeiss Axiophot microscope (camera Zeiss AxioCam MRc5). Images were taken with differential interference contrast (DIC) optics with a 100x objective (*Figure 7B–D*; *Figure 7—figure supplements 1–3*), or dark-field illumination with a 10x objective (*Figure 6A–F*; *Figure 7—figure supplement 2*). For the quantification of denticle numbers (*Figure 7B,C*), near-normal-looking abdominal denticle belts were chosen under dark-field, and denticle numbers were counted at high magnification; for *Figure 7B*, belts from abdominal segments 3–6 were chosen.

## Antibody staining of embryos

Dechorionated embryos were washed in water and fixed by shaking in 1:3 phosphate-buffered saline (PBS) and heptane containing 4% formaldehyde for 30 min. Ice-cold methanol was added, and tubes were vortexed for 1 min. Embryos were washed in methanol 5x and rehydrated by 5 min washes in 90, 75, 50, and 25% of methanol followed by a final PBS wash. Embryos were permeabilized in PBT (PBS containing 0.1% Triton X-100) for 30 min, blocked in BBT (1% BSA in PBT) for 1 hr and incubated with primary antibodies (α-β-gal 1:500; α-Wg 1:10; α-En 1:10) overnight at 4°C. After washing (3x in BBT), embryos were incubated with biotinylated goat-anti-mouse IgG antibody (1:100 in BBT) for 4 hr and washed (2x in BBT, 1x in PBS). Detection was carried out with ABC Elite kit according to the manufacturer's protocol. Homozygous $nkd^{13}$ embryos were identified by the lack of $twi.lacZ$-bearing balancer chromosomes.

## Mass spectrometry

Mass spectrometry was performed by the LMB Mass Spectrometry Facility. Briefly, peptides from in situ trypsin digestion were extracted in 2% formic acid/2% acetonitrile mix. Digests were analyzed by nano-scale capillary LC-MS/MS using an Ultimate U3000 HPLC and C18 Acclaim PepMap100 nanoViper (Thermo Scientific Dionex). LC-MS/MS data were searched against a protein database (UniProt KB) with the Mascot search engine program (Matrix Science). MS/MS data were validated using the Scaffold program (Proteome Software). MALDI-TOF mass spectrometric measurements were carried out in positive ion mode on an Ultraflex III TOF-TOF instrument (Bruker Daltonik), using sinapinic acid (Sigma) as matrix.

## Protein expression and purification

Protein-coding sequences were inserted in pET vectors bearing His6-Lip (PDZ: residues 285–373 from human DVL2), His6-RFP (EF-hand: residues 110–190 from human NKD1), Strep-MBP and Strep-Lip (HisC sequences; *Figure 3A*). Proteins were expressed in *E. coli* BL21DE3 pRare2 cells (from Marc Fiedler) overnight at 24°C after induction with IPTG at OD600 of 0.7–0.9. Cells were pelleted for 20 min at 7000 g, resuspended in 30 mM Tris-HCl, 300 mM NaCl, pH 7.4 and lysed with sonication (10 s on, 10 s off, 2 min in total, 90% amplitude). Lysates were cleared by centrifugation at 100 000x *g* for 20 min, and clear supernatant was loaded onto NiNTA beads. Beads were washed with 10 column volumes of 30 mM Tris-HCl, 300 mM NaCl, pH 7.4, and proteins were eluted in the same buffer, supplemented with 300 mM imidazole. Fractions containing protein of interest were pooled and further purified by SEC with a S200 Superdex 26/60 column in 15 mM Tris-HCl, 150 mM NaCl, pH 7.4, or in 15 mM acetate, 150 mM NaCl, pH 5.0.

## ITC

ITC was performed on MicroCal iTC200 in 15 mM Tris-HCl, 150 mM NaCl, pH 7.4 by injecting Lip-DVL2-PDZ or Lip-DVL2-PDZ-V334E (696 µM) into 17.2 µM solution of RFP-NKD-EF (preinjection of 0.5 µl + 19 injections of 2 µl). The data were analyzed following the manufacturer's guidelines.

## SEC-MALS

Purified proteins and preformed complexes were analyzed with an Agilent 1200 Series chromatography system connected to a Dawn Heleos II 18-angle light-scattering detector combined with an Optilab rEX differential refractometer (Wyatt). Samples were loaded onto a Superdex-200 10/300 gel filtration column (GE Healthcare) at 2 mg ml$^{-1}$ and run at 0.5 ml/min in buffer (15 mM TrisHCl, 150 mM NaCl, pH 7.4), and the data were processed with Astra V software.

## Co-aggregation assay

Protein samples were prepared in 15 mM acetate, 150 mM NaCl, pH 5.0 and mixed in different ratios. After 10 min, 1 M Tris-HCl, pH 8.0 was added to a final concentration of 200 mM. After 20 min incubation, LDS sample buffer was added, and samples were boiled for 10 min.

## Quantitation and statistical analysis

All error bars are represented as mean ± SEM from >3 independent experiments. Statistical significance was calculated by one-way ANOVA with multiple comparisons (*Figure 1A*; *Figure 2—figure*

*supplement 1*), two-way ANOVA with multiple comparisons (*Figure 5A,B*; *Figure 5—figure supplement 2*) or Student's t tests (*Figure 7B*; *Figure 1—figure supplement 2*), and denoted as follows: [*]$p < 0.05$, [**]$p < 0.01$,[***]$p < 0.001$, [****]$p < 0.0001$.

## Acknowledgements

We thank Keith Wharton for plasmids, Mark Skehel and his team for mass spectrometry, Maria Daly for cell sorting, Balaji Santhanam for advice on the bioinformatics, Chris Johnson for technical assistance with the biophysics, and Hugh Pelham for discussion. This work was supported by Cancer Research UK (C7379/A24639) and the Medical Research Council (U105192713).

## Additional information

### Funding

| Funder | Grant reference number | Author |
| --- | --- | --- |
| Cancer Research UK | C7379/A24639 | Mariann Bienz |
| Medical Research Council | U105192713 | Mariann Bienz |

The funders had no role in study design, data collection and interpretation, or the decision to submit the work for publication.

### Author contributions

Melissa V Gammons, Joshua E Flack, Formal analysis, Validation, Investigation, Visualization, Writing - review and editing; Miha Renko, Juliusz Mieszczanek, Formal analysis, Validation, Investigation, Visualization, Methodology, Writing - review and editing; Mariann Bienz, Conceptualization, Resources, Supervision, Funding acquisition, Writing - original draft, Writing - review and editing

### Author ORCIDs

Melissa V Gammons (iD) https://orcid.org/0000-0001-9661-9331
Mariann Bienz (iD) https://orcid.org/0000-0002-7170-8706

### Decision letter and Author response

Decision letter https://doi.org/10.7554/eLife.59879.sa1
Author response https://doi.org/10.7554/eLife.59879.sa2

## Additional files

### Supplementary files

- Supplementary file 1. Uncropped Western blots.
- Transparent reporting form

### Data availability

All data generated and analysed during this study are included in the manuscript and supporting files.

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
