## [Decision Letter]

**Acceptance summary:**

This manuscript provides evidence that Naked (Nkd) is a context-dependent regulator of Wnt, which not only downregulates Wnt activity but can also act as an activator. The authors propose, contrary to previous reports, that Nkd is a context-dependent regulator of Wnt, which not only downregulates Wnt activity but can also act as an activator. The manuscript represents a significant contribution to the field and fits well into the scope of *eLife*.

**Decision letter after peer review:**

Thank you for submitting your article "Feedback control of Wnt signaling based on histidine cluster co-aggregation between Naked/NKD and Axin" for consideration by *eLife*. Your article has been reviewed by two peer reviewers, and the evaluation has been overseen by a Reviewing Editor and Philip Cole as the Senior Editor. The following individual involved in review of your submission has agreed to reveal their identity: Stephanie Grainger (Reviewer #3).

The reviewers have discussed the reviews with one another and the Reviewing Editor has drafted this decision to help you prepare a revised submission.

Summary:

In the current manuscript, Gammons et al. provide evidence that Nkd is a context-dependent regulator of Wnt, which not only downregulates Wnt activity but can also act as an activator. The authors propose, contrary to previous evidence, that Nkd is a context-dependent regulator of Wnt, which not only downregulates Wnt activity but can also act as an activator. The authors first show that Nkd can be found in high-molecular weight (HMW) clusters and that this aggregation is dependent on a polyhistidine stretch. This His-C stretch leads to aggregates also when fused to other proteins. A proteome-wide search in the human genome led to the discovery that a similar sequence can be found in Axin1 where it is similar required for aggregation. Nkd is shown in the study to bind to Axin1 both in vitro and in vivo. The authors then went on to delete the His-C stretch in Nkd by CRISPR in human cells, or make a complete Nkd knock-out, and surprisingly showed that Wnt activity in these cells was lower, not high as expected for a negative regulator. The authors then turned to *Drosophila* and through analysis of newly generated alleles demonstrated that Nkd can act as negative or positive regulator in a context-dependent manner during embryogenesis. The authors performed a number of elegant and convincing experiments using in vivo and vitro settings and the manuscript is well written. The manuscript represents a significant contribution to the field and fits well into the scope of the journal. There are no major experimental concerns. We suggest the authors remove the p62 data, as these seem not well integrated. The authors should address the comments below.

– The authors could more critically discuss which evidence has been provided that His-C HMW-clusters are formed in cells or in *Drosophila* under non-overexpression conditions.

– Does *Drosophila* Axn has a similar His-C amino acid cluster which would further support the authors model? If not, what would this mean for the requirement of His-C aggregates for signaling?

– The Introduction is very long and could be shortened. It might be helpful if the authors would include other viewpoints (either in the Introduction or Discussion) on the significant of Dvl high molecular complexes.

– Ma et al., 2020 (Kirschner lab) recent findings on small molecule dynamics of Dishevelled should be discussed and cited

– In the Abstract, "epithelial cells whose HisC has been deleted": It would be clearer if the authors add NKD in front of HisC.

– The authors state that "Thus, Nkd1 appears to be targeted to Axin-containing Wnt signalosomes via simultaneous binding to DVL and HisC-dependent co-aggregation with Axin." How do we know these are signalosomes, rather than unbound Axin or Dsh? We suggest that this statement could be softened.

– The blot in Figure 4B pointing to HMW NKD is of poor quality. Can it be redone? Or perhaps there is another exposure available?

– In the Results, when the authors discuss the DVL2-GFP mutants, the rationale for these particular mutations is not clear. A clarifying statement would aid in the readability of the manuscript.

– Some of the figures are presented out of order. For example, Figure 1—figure supplement 2 is mentioned in the text before Figure 1—figure supplement 1.

– Throughout, the immunoblots should have molecular weight markers, and should be cropped to allow 5 band widths of space above and below.

– The data would be further clarified by adding the cell lines used to the figure legends.

– In Figure 7, demarking the row number at the bottom or the top might aid non-specialists in interpreting the data.

---

## [Author Response]

[…] The manuscript represents a significant contribution to the field and fits well into the scope of the journal. There are no major experimental concerns. We suggest the authors remove the p62 data, as these seem not well integrated. The authors should address the comments below.

We would much prefer to keep the p62 data as part of our study because these data are robust, and striking: p62 and NBR1 were the top hits in our BioID screen and, apart from Axin, the only strongly HisC-dependent hits (Figure 8B). Furthermore, there is a pronounced preference of p62 to interact with aggregated versus low-molecular weight NKD1 (Figure 8C, D), consistent with the general notion that autophagy tends to target aggregated protein (e.g. Ma et al., 2015). Last but not least, our p62 data are highly consistent with previous work by the Chen group showing that Dishevelled is targeted for degradation by autophagy by interacting with p62 and GABARAP (Gao et al., 2010; Zhang et al., 2011). The p62 data allow us to suggest autophagy as a route for Naked/NKD-dependent degradation of Wnt signalosomes during prolonged Wnt signalling, which we hope will stimulate further work to consolidate this notion.

– The authors could more critically discuss which evidence has been provided that His-C HMW-clusters are formed in cells or in Drosophila under non-overexpression conditions.

This could only be established conclusively by antibodies that are sensitive enough to detect endogenous Naked/NKD proteins, but these are unfortunately not available. We now state this explicitly in the last paragraph of the subsection “The Nkd1 HisC is crucial for ternary complex formation with Axin and DVL2”.

– Does Drosophila Axn has a similar His-C amino acid cluster which would further support the authors model? If not, what would this mean for the requirement of His-C aggregates for signaling?

Good point! We should have looked at this earlier: it turns out that *Drosophila* Axin does not have a histidine cluster. In fact, histidine clusters seem to be somewhat patchy amongst invertebrate Axins (and not always present in invertebrate NKD proteins either, as shown in Figure 1—figure supplement 1). We therefore inserted the following paragraph into our Discussion:

“Our preliminary searches have indicated that histidine clusters in NKD and Axin proteins are completely conserved throughout vertebrates, however their occurrence amongst invertebrates are more patchy. […] It is therefore possible that some invertebrate NKD proteins (i) use a different mechanism to co-cluster with Axin, (ii) do not form clusters to associate with Axin, or (iii) do not associate with Axin at all. It will be interesting to explore this in future studies.”

– The Introduction is very long and could be shortened. It might be helpful if the authors would include other viewpoints (either in the Introduction or Discussion) on the significant of Dvl high molecular complexes.

We agree – especially the lengthy fourth paragraph detailing the different functions of NKD in the development of various vertebrates. Since this is only of limited relevance to our study, we have therefore shortened this paragraph substantially (see revised Introduction).

We are not quite sure what is meant by the suggestion to ‘include other viewpoints on the significant of Dvl high molecular complexes’, but assume this to refer to the size of endogenous Dishevelled signalosomes. We have therefore inserted a sentence in the revised Introduction (second paragraph) to state that endogenous signalosomes arise from limited Dishevelled polymerisation, referring to two recent imaging studies (from the Weis and Kirschner labs) that were published around the time of submission of our manuscript (Kan et al., 2020, and Ma et al., 2020).

– Ma et al., 2020 (Kirschner lab) recent findings on small molecule dynamics of Dishevelled should be discussed and cited

Done (see point above). However, we do not think it appropriate to include an extensive discussion on the size of endogenous Wnt signalosomes since this is not directly relevant to our study (note also that neither imaging study cited above looked at Wnt signalosomes after prolonged Wnt stimulation, i.e. under the conditions in which Naked/NKD co-aggregates with Axin).

– In the Abstract, "epithelial cells whose HisC has been deleted": It would be clearer if the authors add NKD in front of HisC.

Done (see revised Abstract).

– The authors state that "Thus, Nkd1 appears to be targeted to Axin-containing Wnt signalosomes via simultaneous binding to DVL and HisC-dependent co-aggregation with Axin." How do we know these are signalosomes, rather than unbound Axin or Dsh? We suggest that this statement could be softened.

We have rephrased this sentence to make it clear that this is merely one suggestion consistent with the data (subsection “The Nkd1 HisC is crucial for ternary complex formation with Axin and DVL2”).

– The blot in Figure 4B pointing to HMW NKD is of poor quality. Can it be redone? Or perhaps there is another exposure available?

Thanks for pointing this out: unfortunately, the image of this blot got stretched accidentally during the preparation of the figure, and so the HMW Nkd1 clusters looked ‘smeary’. We have now corrected this (see revised Figure 4B).

– In the Results, when the authors discuss the DVL2-GFP mutants, the rationale for these particular mutations is not clear. A clarifying statement would aid in the readability of the manuscript.

These mutants were used to test whether the polymerisation or dimerisation of Dishevelled is required for promoting aggregation of Nkd1, which we now state in an introductory sentence of a new paragraph addressing these questions (subsection “Dishevelled promotes co-aggregation between Nkd1 and Axin1”, second paragraph.

– Some of the figures are presented out of order. For example, Figure 1—figure supplement 2 is mentioned in the text before Figure 1—figure supplement 1.

This is not the case: we refer to Figure 1—figure supplement 1 in the Introduction, before our first referral to Figure 1—figure supplement 2 in the Results. We have also double-checked that all the other figures (and panels) are cited in order throughout the main text.

– Throughout, the immunoblots should have molecular weight markers, and should be cropped to allow 5 band widths of space above and below.

We have added molecular weight markers to the most crucial first blot showing LMW and HMW Nkd1 (Figure 1C), which then applies to all subsequent blots which we have cropped to show only slices containing the relevant bands. Labelling all these slices with just one marker (on none) seems somewhat pointless, and increasing their widths (by a factor of 2-3, as suggested) would make most of our figures rather large and unwieldy. Instead, we suggest that we deposit all uncropped original blots as Supplementary file 1, which should address both suggestions as well as provide raw data information.

– The data would be further clarified by adding the cell lines used to the figure legends.

Done (see revised legends for Figures 1, 4, 5 and 8).

– In Figure 7, demarking the row number at the bottom or the top might aid non-specialists in interpreting the data.

Done (see revised Figure 7B where we have labelled the denticle rows in the wt cuticle).